# Treatment Intensification Prior to Radical Prostatectomy for Clinically Localized Prostate Cancer

**DOI:** 10.3390/cancers17132258

**Published:** 2025-07-07

**Authors:** Carlos Jesus Perez Kerkvliet, Joon Yau Leong, Rasheed A. M. Thompson, Kevin Kayvan Zarrabi, William Kevin Kelly, Costas Lallas, Leonard Gomella, Mihir Shah

**Affiliations:** 1Department of Urology, Sidney Kimmel Cancer Center, Thomas Jefferson University Hospital, Philadelphia, PA 19107, USA; carlos.perez2@jefferson.edu (C.J.P.K.); joonyau.leong@jefferson.edu (J.Y.L.); rasheed.thompson@jefferson.edu (R.A.M.T.); costas.lallas@jefferson.edu (C.L.); leonard.gomella@jefferson.edu (L.G.); 2Department of Medical Oncology, Sidney Kimmel Cancer Center, Thomas Jefferson University Hospital, Philadelphia, PA 10107, USA; kevin.zarrabi@jefferson.edu (K.K.Z.); william.kelly@jefferson.edu (W.K.K.)

**Keywords:** prostate, neoadjuvant, chemotherapy, cancer, prostatectomy

## Abstract

For unfavorable intermediate- or high-risk prostate cancer, current guidelines primarily recommend radical prostatectomy (RP) or radiation with androgen deprivation therapy (ADT). While emerging interest exists in neoadjuvant ADT prior to RP for this population, previous trials, despite suggesting benefits like reduced surgical complexity, pathologic downstaging, decreased positive margins, and lower nodal positivity, have not demonstrated improvements in cancer progression or survival, thereby precluding its routine recommendation for surgical patients. Conversely, as ADT remains a cornerstone for metastatic disease, there is ongoing exploration of its expanded neoadjuvant use for clinically localized disease, with several ongoing trials investigating second-generation androgen blockers, radiopharmaceuticals, and PARP inhibitors to assess potential long-term cancer-specific survival benefits. This review will comprehensively overview the recent literature and ongoing efforts to integrate neoadjuvant therapy for clinically localized prostate cancer patients at high risk of recurrence post-prostatectomy.

## 1. Introduction

Prostate cancer affects one in eight men during their lifetime. Ultimately, 1 out of 44 men will die of prostate cancer [1]. Over the last few decades, it has been recognized that the wide spectrum of severity and aggressiveness of prostate cancer is dictated by the molecular underpinnings encompassed within each grade group [2]. As such, with the advancement in the understanding of the biology of prostate cancer, we have adjusted our therapeutic approach. Various efforts have been made through guideline recommendations to better establish ways to treat prostate cancer based on risk stratification [3,4]. In some instances, efforts have been to “de-intensify” the therapeutic management of certain prostate cancer subgroups and perhaps encourage the concept of active surveillance [4]. However, there are also efforts to intensify our approach for patients with high-risk and very high-risk disease. Currently, guidelines provided by the National Comprehensive Cancer Network (NCCN) and American Urological Association (AUA) both strongly discourage clinicians from utilizing neoadjuvant therapy for clinically localized disease prior to radical prostatectomy [3,4]. However, there is a growing body of evidence that points towards a potential need to further explore and consider these avenues, especially in patients with high-risk localized prostate cancer. In this paper, we aim to describe and summarize the evidence supporting neoadjuvant therapies which might be useful in clinically localized high-risk patients who will undergo radical prostatectomy.

## 2. Rationale and Biological Principle

Neoadjuvant therapies have been successfully utilized in many solid tumor malignancies [5,6,7,8]. Specifically, within the discipline of urology, neoadjuvant chemotherapy has had a well-defined role in bladder cancer and upper tract urothelial carcinoma, where studies have demonstrated that the progression-free survival and overall survival is improved among patients who are treated with neoadjuvant chemotherapy and later undergo surgery [9,10].

It is likely that the agents utilized as part of the treatment for the adjuvant therapeutic approach after radical prostatectomy can also be used as neoadjuvant agents. This is especially the case since there are studies that have already established a solid role of radical prostatectomy in men who have oligometastatic disease in molecular imaging such as PSMA. Thus, there is an increase in using neoadjuvant drugs prior to radical prostatectomy [11]. Specifically, this approach will be useful in patients who will likely need these types of therapies in the future, such as high-risk and very high-risk prostate cancer patients. Although there are different definitions for high-risk and very high-risk prostate cancer, most of the patients are typically defined as having a PSA of >20, being in grade group 4 or above, and having clinical stage T3 disease [3,4]. In contemporary practice, patients who are considered high-risk or very high-risk are recommended to undergo imaging studies to assess clinical staging. This imaging is performed in the hope of better selecting the patient who will likely benefit from prostatectomy and the patient who might be a better fit to proceed with a guideline-oriented alternative therapy such as radiation therapy in conjugation with ADT or chemotherapy. However, it is important to establish that the decision of proceeding with surgery for patients with high-risk or very high-risk prostate cancer does not solely rely on imaging. In many academic centers, these decisions are often guided by the discussions of practitioners on tumor boards [3]. Anecdotally, in our center, many nodal diseases or metastatic diseases to the bone would deter practitioners from proceeding with a prostatectomy. However, no data has yet demonstrated the potential use of neoadjuvant systemic therapies to prompt certain patients, with perhaps locoregional disease, to be better candidates for prostatectomy.

Similar to other cancer subtypes driven by hormones, prostate growth can be deterred by the use of androgen ablation [12]. Specifically, the androgen hormonal pathway presents a unique opportunity to regress local, regional, and micro metastatic diseases [13]. Thus, contrary to other urological malignancies, prostate cancer management with ADT poses a unique opportunity to decrease androgen-driven molecular pathways and further decrease tumoral size and micro metastasis.

In recent years, capitalizing on the immunogenicity of solid tumor malignancies, novel immunotherapies have been evaluated in the neoadjuvant setting [14,15]. Unfortunately, the tumor mutational burden in prostate cancer is often low and the tumor microenvironment harbors an immune milieu that is prohibitive to robust immune anti-tumor responses [16]. This likely limits the utility of immunotherapeutics as an efficacious approach to prostate cancer [17]. However, there may be a subset of patients that may benefit from the use of these agents, especially if there is a high mutational tumor burden [16]. Another treatment avenue that has been utilized in prostate cancer is chemotherapeutic agents, which are currently mainly used in the adjuvant setting [3,4]. This might also represent the possibility of potentially combining employment with ADT or immunotherapeutics.

Lastly, there is an increasing use of antigen radioligands in the setting of metastatic castration-resistant prostate cancer [18]. Most likely its use could be expanded to the neoadjuvant setting as well, with the goal of decreasing tumor size or even the regression of extraprostatic extension prior to a radical prostatectomy. Herein, in this paper, we will describe the clinical trials and research endeavors associated with these four potential therapeutic approaches that can be utilized in the neoadjuvant setting for high-risk and very high-risk prostate cancer patients: hormonal therapy, chemotherapeutics, immunotherapy, and radioligands.

## 3. Hormonal Therapy

Prostate cancer is one of the few types of cancer in which steroid hormone receptors provide a unique opportunity for the management of cellular mechanisms associated with carcinogenesis [19]. It is well known that hormonal deprivation leads to the inhibition of cancer growth as well as potentially decreases micro-metastatic lesions [20]. Androgen signaling inhibition in prostate cancer is achieved by two primary methods: (1) by decreasing the androgen production, and (2) by inhibiting and modulating the activity of the androgen receptor intratumorally [21].

There are several seminal studies exploring the actions of androgen deprivation in the neoadjuvant setting. First, Labrie et al. demonstrated that cancer positive margins were reduced from 38.5% in control patients to 13% in men who received a neoadjuvant combination of leuprolide and flutamide [22]. This finding was similarly confirmed by Gravina et al. in 2007 when they demonstrated that neoadjuvant treatment with bicalutamide reduced the positive surgical margins in men with T2 and T3a prostate cancer as well, with a reduction of 13.1% in the bicalutamide group vs. 34.5% in the control group [23]. Interestingly, although there are several studies demonstrating efficacy with regard to the decrease in positive margins, there are also studies that found no difference in the risk of biochemical recurrence-free survival despite a margin decrease when using neoadjuvant cyproterone [24,25]. However, the ARNEO trial might have the potential to further explore the avenue of androgen deprivation in the neoadjuvant setting. Specifically, the ARNEO trial evaluated neoadjuvant degarelix with or without apalutamide before radical prostatectomy in high-risk prostate cancer patients. This phase II trial showed that degarelix together with apalutamide resulted in improved pathological responses, specifically higher rates of minimal residual disease. However, longer-term follow-up revealed no significant difference in the biochemical recurrence between the treatment arms. Notably, achieving organ-confined disease (ypT2) post-treatment was associated with better biochemical recurrence-free survival. The trial suggests that while the combination impacts pathological outcomes, long-term recurrence benefits require further study [26,27].

The studies described above employed only the use of ADT and not the newer generation antiandrogen therapies that have been employed in recent years [28,29]. Importantly, these novel antiandrogen agents can specifically decrease the transcriptional activity of the androgen receptor via multiple mechanisms that have been described in the literature [30,31]. Firstly, at times, they can decrease the expression of the androgen receptor. Secondly, they can decrease the activity of the androgen receptor itself by binding as an inhibitor ligand. Lastly, these agents can also decrease the transcriptional activity of the androgen receptor, which, ultimately, is the main way the androgen receptor employs its carcinogenic activities in prostate cancer [31]. Because of these features and the intratumoral potential activity that these newer agents would have, one would expect that the neoadjuvant efficacy of these drugs could potentially be even better than that of the androgen deprivation drugs.

Unfortunately, studies that have explored the neoadjuvant role of these new agents as monotherapies to decrease the progression of prostate cancer have failed to demonstrate any significant survival benefit. Montgomery et al. demonstrated that patients who underwent prostatectomy after six months of enzalutamide with dutastaride and a luteinizing hormone releasing hormone analog, compared with enzalutamide monotherapy, had better pathologic complete response and lower residual cancer burden [32]. The study also suggested that neoadjuvant monotherapy with enzalutamide may not be sufficient for clinical benefit, indicating the potential need for combination therapies [32]. McCay et al. assessed the addition of abiraterone to enzalutamide and LHRH agonist, showing a trend towards better outcomes, though not statistically significant [33]. Notably, Ravi et al. analyzed data from five clinical trials involving patients with high-risk localized prostate cancer who received intensified neoadjuvant therapy with androgen deprivation therapy and an androgen receptor pathway inhibitor before radical prostatectomy. This study found that the extent of residual disease at radical prostatectomy, measured as residual cancer burden, was prognostic for metastasis-free survival. Specifically, patients with lower residual cancer burden had significantly higher 5-year metastasis-free survival rates. The authors suggested that residual cancer burden could be a valuable tool in guiding future neoadjuvant therapies and post-neoadjuvant therapies in trials for high-risk localized prostate cancer [34].

Taken together, these data suggest that novel antiandrogen monotherapy has suboptimal castration activity and unfortunately, is unable to reach any clinical benefit in the neoadjuvant setting. More importantly, even combinatorial approaches with antiandrogen agents have failed to demonstrate robust effects in dampening prostate cancer progression. Interestingly, the androgen deprivation approaches have better outcomes. One can hypothesize that this could be due to effects of androgen deprivation in other cell types within the tumor that perhaps benefit from pro-androgenic activities despite androgen receptor overdrive. More importantly, some studies have pointed out that despite the histopathologic response of the cancer cells themselves (i.e., surgical margins), there are no effects in biochemical recurrence. This could again be due to the fact that some of these agents only target the activity of the androgen receptor in prostate cancer cells without, at times, having an influence in the surrounding tumor milieu. As above, more studies are needed to better define which patients would benefit from these therapeutics. At this time, it appears that residual cancer burden might be what can be employed to better identify patients who would benefit from neoadjuvant hormonal therapy.

## 4. Chemotherapeutics

Although the use of chemotherapeutics has been amply explored in the neoadjuvant setting of prostate cancer, there is not a lot of data about the use of chemotherapeutics as monotherapy. However, there are some studies that have explored adding chemotherapeutic agents to hormonal therapy in the adjuvant setting. Using chemotherapeutic agents, in addition to the hormonal approach, has the advantage of targeting cells that circumvent and are not dependent on androgen receptor activity. For instance, a Phase II trial by Chi et al. involved 72 high-risk patients who received ADT and docetaxel before prostatectomy, resulting in a 30% relapse rate at a median follow-up of 42.7 months [35]. More importantly, the PUNCH Alliance 90203 trial compared radical prostatectomy alone to radical prostatectomy with six cycles of docetaxel and an LHRH analog over 18–24 weeks. The study included 788 patients with localized high-risk prostate cancer. Results showed that those receiving the combined therapy had lower pathologic stages, fewer positive surgical margins, and fewer metastatic lymph nodes at the time of surgery. Although the trial did not achieve its primary goal of reducing biochemical recurrence (BCR) at three years, longer-term follow-up pointed towards improved BCR-free survival and overall survival rates [36]. Lastly, the CALGB study had demonstrated that chemotherapy together with targeted therapy could be superior to chemotherapy alone [36,37]. This further strengthens the hypothesis that although chemotherapy can be used in the neoadjuvant setting, it is likely that some other agents are needed to better achieve its benefits in patients who are to undergo definitive management of prostate cancer.

PARP (Poly (ADP-ribose) polymerase) inhibitors represent another possible treatment for prostate cancer in the neoadjuvant setting. However, there are currently no studies that have explored the role of PARP inhibitors in neoadjuvant prostate cancer management, although some efforts are ongoing [38]. Importantly, PARP inhibitors, such as Olaparib, have been proven to have good effects when combined with androgen receptor modulators in metastatic hormone-sensitive prostate cancer [39]. More importantly, they have recently been found to have a role in metastatic castration-resistant prostate cancer as well [40], thus indicating that these agents could have a role regardless of the androgen receptor status and dependence of the cancer cells. Given these results, it is possible that these inhibitors could be utilized as a neoadjuvant approach with good results for high-risk and very high-risk prostate cancer. Further studies are needed to evaluate the role of this therapeutic approach prior to prostatectomy.

## 5. Immunotherapy

Prostate cancer is not a traditional cancer subtype that has an ample amount of tumor mutational burden [41]. Interestingly as well, within the tissue microenviroment of the prostate, there is an abundance of regulatory immune cells, such as M2 macrophages as well as regulatory T-cells, that would dampen the immune response within the prostatic tissue. However, in recent studies, there have been multiple approaches to exploit some avenues that directly potentiate an immune reaction against prostate tumors. Notably, there has been a lot of interest with regard to inducing the activity of CD3 T cells by binding to a transmembrane protein metalloreductase called STEAP1. Specifically, there are studies investigating the action of AMG 509, which is a bispecific T-cell engager that promotes the activity of CD3 T cells against cells that express STEAP1 and is commonly overexpressed in prostate cancer tissue. Interim results from a Phase 1 clinical trial of patients with metastatic castration-resistant prostate cancer refractory to prior novel hormonal therapy and 1–2 taxane regimens has found significant anti-tumor activity, with notable declines in prostate-specific antigen levels and partial responses in some patients [42]. More importantly, pharmacokinetics showed a dose-proportional increase in drug exposure. Although this data is specifically for metastatic castration-resistant prostate cancer, one can evaluate the possibility of using this immunotherapeutic approach in high-risk disease patients who would likely have biological activity that is like the metastatic castration-resistant prostate cancer profile.

Other immunotherapeutic strategies have been used for prostate cancer. Specifically, there have been efforts to evaluate the ability of poxvirus-based vaccines (PROSTVAC-V/F) to elicit an immune response against prostate-specific antigens. Unfortunately, this approach was rendered to be ineffective and did not have any specific improvement in overall survival [43]. Lastly, the effectiveness of checkpoint inhibitors in prostate cancer, such as PD-L1, PD1, and CTLA-4 inhibitors, have been very poor despite multiple attempts to exploit these agents in prostate cancer. This is likely due to the low tumor mutational burden of prostate cancer. Interestingly, recent data has demonstrated some effectiveness in patients with metastatic castration-resistant prostate cancer regardless of the mutational status of the DNA repair defects gene. In fact, the NEPI trial is investigating neoadjuvant LuPSMA and ADT, with or without ipilimumab, in patients with very high-risk prostate cancer, before radical prostatectomy. This phase I/II study primarily assesses the feasibility and safety of this combination, especially when adding ipilimumab. The primary endpoints are the feasibility of subsequent surgery and the rate of complete pathologic response. The trial aims to determine if this combined approach enhances tumor response in this high-risk group [44]. Trials such as this one will assist in determining how to better classify patients for intensification of management prior to radical prostatectomy. Specifically, better biomarkers to improve predictions of which patients might benefit from PD-L1, PD1 and CTLA-4 blockers are needed [45].

## 6. Future Steps

As discussed in prior sections in this paper, there are multiple approaches to the neoadjuvant management of prostate cancer. In an attempt to describe each of the approaches, Table 1 summarizes some of the most important clinical trials that are currently ongoing. Interestingly, there is another unexplored aspect in the field of neoadjuvant chemotherapeutics that we believe should be further studied and that is the role of neoadjuvant radiation therapy. Unfortunately, although this might represent an avenue that could be explored, the challenge comes when performing prostatectomy after [46], which is technically very difficult. Also, focal therapy in prostate cancer is an emerging field for the management and treatment of prostate cancer [47]. More research is needed with regard to the role of neoadjuvant therapies in the context of focal therapy. Lastly, it is known that patients are extremely important in guiding the management of the therapeutic approach in their disease [48]. It would be very interesting to investigate how the role of neoadjuvant approaches in prostate cancer prior to radical prostatectomy will intertwine with patients’ preferences and priorities.

## 7. Conclusions

Neoadjuvant therapy for high-risk, localized prostate cancer represents a promising avenue for improving patient outcomes. While hormonal therapy has shown mixed results, the combination of chemotherapeutics and hormonal therapy has demonstrated potential in reducing disease burden and improving survival. Immunotherapy, while still in its early stages, offers a potential new approach, particularly for patients with high tumor mutational burdens. Further research and clinical trials are needed, and ongoing as well, to fully elucidate the benefits and risks of neoadjuvant therapy in this patient population and to establish optimal treatment regimens. As our understanding of prostate cancer biology continues to evolve, neoadjuvant therapy may play an increasingly important role in the management of high-risk, localized disease.

## Figures and Tables

**Table 1 cancers-17-02258-t001:** Ongoing clinical trials with regard to neoadjuvant chemotherapeutic approaches.

NCT Number	Study Title	Planned Interventions	Phases
NCT06259123	Neoadjuvant PSMA-RLT in Oligometastatic PCa	PSMA	Phase 2
NCT06387056	Genomic Biomarker-guided Neoadjuvant Therapy for Prostate Cancer (SEGNO)	Rezvilutamide, Goserelin Microspheres for Injection, Docetaxel, Pamiparib, Cisplatin, Tislelizumab	Phase 2
NCT04887935	Neoadjuvant SGLT2 Inhibition in High-Risk Localized Prostate Cancer	Dapagliflozin	Phase 1
NCT06575257	Neoadjuvant Therapy of Darolutamide Plus ADT for High-Risk Prostate Cancer	Darolutamide, Goserelin 3.6 mg	Phase 2
NCT06631521	Neoadjuvant Darolutamide and Relugolix Combination Preceding Radical Prostatectomy for Prostate Cancer	Darolutamide, Relugolix	Phase 1
NCT05223582	Fluzoparib and Abiraterone in the preSurgery Treatment of Prostate Cancer: FAST Trial	Abiraterone acetate, Fluzoparib, Prednisone	Phase 2
NCT06613100	Evaluation of Neoadjuvant Xaluritamig in Localized Prostate Cancer	Xaluritamig	Phase 1
NCT03821246	Neoadjuvant Atezolizumab-Based Combination Therapy in Men with Localized Prostate Cancer Prior to Radical Prostatectomy	Atezolizumab, Tocilizumab, Etrumadenant	Phase 2
NCT04894188	Neoadjuvant Hormone and Radiation Therapy Followed by Radical Prostatectomy in Patients with High-Risk Prostate Cancer	Goserelin 3.6 mg	NA
NCT02903368	Neoadjuvant And Adjuvant Abiraterone Acetate + Apalutamide Prostate Cancer Undergoing Prostatectomy	Apalutamide, Leuprolide, Prednisone, Abiraterone Acetate	Phase 2
NCT06347705	A Study of 2141-V11 in Combination with Standard Treatments in People with Prostate Cancer	2141-V11 Antibody	Phase 2
NCT05593497	A Single-Arm Phase II Study of Neoadjuvant Intensified Androgen Deprivation (Leuprolide and Abiraterone Acetate) in Combination with AKT Inhibition (Capivasertib) for High-Risk Localized Prostate Cancer with PTEN Loss	Capivasertib, abiraterone acetate	Phase 2
NCT05406999	Neoadjuvant Intense Endocrine Therapy for High Risk and Locally Advanced Prostate Cancer	Abiraterone Acetate, Prednisolone, Enzalutamide, Apalutamide, Darotamide, Rezvilutamide	Phase 2
NCT05249712	Efficacy and Safety of Darolutamide Combined with ADT in High-risk/Very High-risk Localized Prostate Cancer	Darolutamide	Phase 2
NCT06066437	Neoadjuvant Theranostic Lutetium Study: The Nautilus Trial	177Lu rhPSMA-10.1, Deguelin	Phase 2
NCT06029036	A Phase II Neoadjuvant Study of Darolutamide Plus ADT in Men with Localized Prostate Cancer	Darolutamide+ADT	Phase 2
NCT03080116	Neoadjuvant Degarelix with or Without Apalutamide (ARN-509) Followed by Radical Prostatectomy	ARN-509, Degarelix	Phase 2
NCT06014255	Trial of Neoadjuvant Enoblituzumab vs. SOC in Men with High-Risk Localized Prostate Cancer	Enoblituzumab	Phase 2
NCT02923180	Neoadjuvant Enoblituzumab (MGA271) in Men with Localized Intermediate and High-Risk Prostate Cancer	Enoblituzumab	Phase 2
NCT01990196	Neoadjuvant Phase 2 Study Comparing the Effects of AR Inhibition With/Without SRC or MEK Inhibition in Prostate Cancer	degarelix, enzalutamide, trametinib, dasatinib	Phase 2
NCT05740488	Efficacy and Safety of Apalutamide in Combination With 89Sr as Neoadjuvant Therapy in Prostate Cancer With‚ Bone Metastases	Apalutamide	NA
NCT04009967	Biomarkers for Neoadjuvant Pembrolizumab in Non-Metastatic Prostate Cancer Positive by 18FDG-PET Scanning	Pembrolizumab	Phase 2
NCT05496959	177-Lutetium-PSMA Before Stereotactic Body Radiotherapy for the Treatment of Oligorecurrent Prostate Cancer, The LUNAR Study	Lutetium Lu-177 PNT2002	Phase 2
NCT05617885	Neo-DAB: Darolutamide and Abemaciclib in Prostate Cancer	Darolutamide, Abemaciclib, Leuprolide, Goserelin, Degarelix	Phase 2
NCT04301414	Non-fucosylated Anti-CTLA-4 (BMS-986218) + Degarelix Acetate vs. Degarelix Acetate Alone in Men with High-risk Localized Prostate Cancer	BMS-986218, Degarelix	Phase 1
NCT04997252	An Evaluation of the Efficacy and Safety of Apalutamide as Neoadjuvant Endocrine Therapy in High-Risk and Oligometastatic Prostate Cancer	Apalutamide	NA
NCT06306612	CytoREductive prostAtectomy for Poly-metastatic Hormone sensiTIVE Prostate Cancer	Systemic Chemohormonal Therapy, Systemic Hormonal Therapy	NA
NCT03258320	A Randomised Trial of Cabazitaxel, Docetaxel, Mitoxantrone or Satraplatin (CDMS) Plus Surgery for Prostate Cancer Patients Without Metastasis	Cabazitaxel, Docetaxel, Mitoxantrone or Satraplatin	Phase 1
NCT03860987	Neoadjuvant Androgen Deprivation Therapy Combined with Enzalutamide and Abiraterone Using Multiparametric MRI and 18FDCFPyL PET/CT in Newly Diagnosed Prostate Cancer	Goserelin, Enzalutamide, Abiraterone, 18F-DCFPyL	Phase 2
NCT00329043	Sunitinib Malate With Hormonal Ablation for Patients Who Will Have Prostatectomy	Sunitinib Malate	Phase 2
NCT03412396	Apalutamide in Treating Patients with Prostate Cancer Before Radical Prostatectomy	Apalutamide	Phase 2
NCT05498272	Study of Neoadjuvant PARP Inhibition Followed by Radical Prostatectomy in Patients with Unfavorable Intermediate-Risk or High-Risk Prostate Cancer with Select HRR Gene Alterations	Olaparib	Phase 2
NCT00430183	Surgery With or Without Docetaxel and Leuprolide or Goserelin in Treating Patients with High-Risk Localized Prostate Cancer	Docetaxel	Phase 3
NCT02949284	Androgen Receptor Antagonist ARN-509 With or Without Abiraterone Acetate, Gonadotropin-Releasing Hormone Analog, and Prednisone in Treating Patients with High-Risk Prostate Cancer Undergoing Surgery	Abiraterone Acetate, ARN-509	Phase 2
NCT04030559	Niraparib Before Surgery in Treating Patients with High Risk Localized Prostate Cancer and DNA Damage Response Defects	Niraparib	Phase 2

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
