# Peer review of "Treatment Intensification Prior to Radical Prostatectomy for Clinically Localized Prostate Cancer"

_cancers, 2025, doi:10.3390/cancers17132258_

Round 1
Reviewer 1 Report
Comments and Suggestions for Authors
general comments
In this study, the authors review the effectiveness of neoadjuvant therapy in radical prostatectomy for localized prostate cancer. This is very interesting as neoadjuvant therapy may be useful as a treatment for high-risk or very high-risk prostate cancer. There are several reviews on similar topics, but this one provides relatively detailed information on immunotherapy and radioligands as neoadjuvant therapy, which is likely to be of value. However, there are several issues that need to be addressed:
Specific comments for revision:
- You seem to have not described when and in what databases the literature search for this narrative review was conducted? It may be necessary to describe the methodology and demonstrate that an appropriate literature search was performed.
- In lines 110-114, the authors cite reference 23 to state that despite the reduced margin positivity rate, there was no significant difference in biochemical recurrence. Reference 23 appears to be a paper that describes the pathological effects, so perhaps it does not show that there is no difference in biochemical recurrence? I suggest citing additional references.
- Why is there no significant contribution of preoperative adjuvant hormone therapy to non-recurrence rates despite improving histopathologic outcomes? If you have a discussion of the reasons for this, please describe.
- If you do not need the "Th" on line 159, delete it.
Author Response
We thank Reviewer 1 for the inputs to our paper and their time invested in making this a stronger paper. Here we will address each of the comments.
- You seem to have not described when and in what databases the literature search for this narrative review was conducted? It may be necessary to describe the methodology and demonstrate that an appropriate literature search was performed.
The literature review was conducted using the pubmed database for literature. We did not specifically perform a meta-analytic study and the purpose of this study is rather to offer a comprehensive report of possible therapeutic options for the neoadjuvant setting. Our study is more to provide a possible foundation for future research rather than to specifically provide an analytical and quantifiable framework. We will specifically detailed this in the author contributions section.
- In lines 110-114, the authors cite reference 23 to state that despite the reduced margin positivity rate, there was no significant difference in biochemical recurrence. Reference 23 appears to be a paper that describes the pathological effects, so perhaps it does not show that there is no difference in biochemical recurrence? I suggest citing additional references.
Thank you for noticing this. We have added a reference that now address this and that is no difference in biochemical recurrence and biochemical recurrence free survival when using neoadjuvant hormonal therapy.
- Why is there no significant contribution of preoperative adjuvant hormone therapy to non-recurrence rates despite improving histopathologic outcomes? If you have a discussion of the reasons for this, please describe.
We have added a commentary in lines 177-179 to address this concern. Thank you.
- If you do not need the "Th" on line 159, delete it.
Thank you, we have deleted this.
Reviewer 2 Report
Comments and Suggestions for Authors
The paper is a narrative review about neoadjuvant therapy prior to radical prostatectomy in high-risk prostate cancer. The topic is urological conundrum since the introduction of castration therapies. Despite the lack of evidence, recent therapeutical advancements may play a role. There are many candidates that are being tested in ongoing trials. Combination of drugs seem to be a promising approach. On top of the review, a brief discussion about the role of radical prostatectomy in oligo metastatic patients would help to understand why it is so difficult to find evidence in this clinical setting.
Author Response
We are thankful that reviewer 2 have read our manuscript and has provided very insightful to our manuscript. Here is how we addressed his/her comments.
The paper is a narrative review about neoadjuvant therapy prior to radical prostatectomy in high-risk prostate cancer. The topic is urological conundrum since the introduction of castration therapies. Despite the lack of evidence, recent therapeutical advancements may play a role. There are many candidates that are being tested in ongoing trials. Combination of drugs seem to be a promising approach. On top of the review, a brief discussion about the role of radical prostatectomy in oligo metastatic patients would help to understand why it is so difficult to find evidence in this clinical setting.
We have added a brief discussion about radical prostatectomy in oligometastatic disease in lines 69-70.
Reviewer 3 Report
Comments and Suggestions for Authors
In this manuscript, authors described an actual topic on treatment intensification prior to radical prostatectomy for clinically localized prostate cancer (PCa) Manuscript is interesting however some points should be clarified.
1) What about RT and BT and neoadjuvant therapies in PCa? I would add a brief report on topic too. See and add this reference: Slevin F et al. A Systematic Review of the Efficacy and Toxicity of Brachytherapy Boost Combined with External Beam Radiotherapy for Nonmetastatic Prostate Cancer. Eur Urol Oncol. 2024 Aug;7(4):677-696. doi: 10.1016/j.euo.2023.11.018. Epub 2023 Dec
26. PMID: 38151440
2) What about focal therapy? Shah TT et al. Focal therapy compared to radical prostatectomy for non-metastatic prostate cancer: a propensity score-matched study. Prostate Cancer Prostatic Dis. 2021 Jun;24(2):567-574. doi: 10.1038/s41391-020-00315-y. Epub 2021 Jan 28. PMID: 33504940
3) Finally: what about patients’ preferences? I would add a brief comment on this actual topic. See and add this reference. Guercio A et al. Patient satisfaction and decision regret in patients undergoing radical prostatectomy: a multicenter analysis. Int Urol Nephrol. 2025 Apr 17. doi:
10.1007/s11255-025-04510-5. PMID: 40246765
Author Response
We thank reviewer 3 for their contributions to improving and making this paper stronger.
1) What about RT and BT and neoadjuvant therapies in PCa? I would add a brief report on topic too. See and add this reference: Slevin F et al. A Systematic Review of the Efficacy and Toxicity of Brachytherapy Boost Combined with External Beam Radiotherapy for Nonmetastatic Prostate Cancer. Eur Urol Oncol. 2024 Aug;7(4):677-696. doi: 10.1016/j.euo.2023.11.018. Epub 2023 Dec
26. PMID: 38151440
We have added this paper in our discussion of our paper as well as brief discussion with regards to RT and BT.
2) What about focal therapy? Shah TT et al. Focal therapy compared to radical prostatectomy for non-metastatic prostate cancer: a propensity score-matched study. Prostate Cancer Prostatic Dis. 2021 Jun;24(2):567-574. doi: 10.1038/s41391-020-00315-y. Epub 2021 Jan 28. PMID: 33504940
3) Finally: what about patients’ preferences? I would add a brief comment on this actual topic. See and add this reference. Guercio A et al. Patient satisfaction and decision regret in patients undergoing radical prostatectomy: a multicenter analysis. Int Urol Nephrol. 2025 Apr 17. doi: 10.1007/s11255-025-04510-5. PMID: 40246765
We have addressed both of these in our future steps discussion. We have also added these two references as above.